# Assessment of the Yam Landraces (*Dioscorea* spp.) of DR Congo for Reactions to Pathological Diseases, Yield Potential, and Tuber Quality Characteristics

**Idris I. Adejumobi** [1,2]ⓘ, **Paterne A. Agre** [2,*]ⓘ, **Didy O. Onautshu** [1], **Joseph G. Adheka** [1], **Inacio M. Cipriano** [1]ⓘ, **Jean-Claude L. Monzenga** [3] **and Joseph L. Komoy** [1]

[1]  Department of Biotechnology, Faculty of Science, University of Kisangani, Kisangani 2012, Congo; i.adejumobi@cgiar.org (I.I.A.); didy.onautshu@unikis.ac.cd (D.O.O.); jadheka@yahoo.fr (J.G.A.); inacio.cipriano@uem.ac.mz (I.M.C.); josephkomoy@gmail.com (J.L.K.)

[2]  International Institute of Tropical Agriculture, Ibadan 5320, Nigeria

[3]  Institut Facultaire des Sciences Agronomiques de Yangambi, Kisangani 1232, Congo; claumonz@yahoo.fr

*  Correspondence: p.agre@cgiar.org

**Abstract:** Yams (*Dioscorea* spp.) possess the potential to contribute to food security and poverty alleviation in DR Congo; however, yam production is limited by several constraints, including the lack of yam improvement programs to address challenges relating to yield improvement, resistance to foliar diseases, and post-harvest tuber quality. Identification of a superior genotype for these traits and reservoirs of genes for improvement would guide yams' improvement. This study aims to evaluate and identify landraces with superior performance for farmers and consumers. We evaluated 191 accessions from six yam species, and significant variation in the performances was observed at $p < 0.05$. Accessions of *D. alata* were superior for tuber oxidative browning ($-0.01$), *D. cayenensis* for high yield potential (29 t/ha), *D. bulbifera* for yam mosaic virus (YMV) tolerance (AUDPC = 3.88), and *D. rotundata* for tuber dry matter content (37%). A high genotypic and phenotypic coefficient of variation (>40) was observed for tuber yield, number of tubers per plots, tuber flesh oxidative browning, and tuber flesh texture. High broad-sense heritability estimates (>60) were similarly observed for all the assessed parameters except number of tubers per plot. Tuber size was identified as the best predictor for tuber yield (b = 2.64, $p < 0.001$) and tuber dry matter content (b = 2.21, $p < 0.001$). The study identified twenty stable landrace accessions from three *Dioscorea* species (*D. alata* (7); *D. cayenensis* (2); *D. rotundata* (11)). These accessions combined high yield potential, high tuber dry matter, high tolerance to YMV and YAD, and low tuber flesh oxidation. The accessions could be considered for the establishment of a yam improvement program in DR Congo.

**Keywords:** *D. alata*; *D. bulbifera*; *D. cayenensis*; *D. dumetorum*; *D. praehensilis*; *D. rotundata*; landraces

## 1. Introduction

Root and tuber crops make a significant contribution to global dietary needs after cereal crops [1]. Yam is among the principal root and tuber crops, including cassava and potato, that are widely grown and consumed as subsistence staples [2]. Yam is a generic name for the Dioscorea species widely cultivated in the tropics and subtropics by smallholder farmers mainly for its starchy underground tuber and aerial bulbils [3,4]. Thus, yam is a group of economically important multi-species crops that serve as a valuable source of food across Africa, Asia, South America, the Caribbean, and the Pacific [5,6]. The global estimated mean annually for yam production and gross values are approximately 73 million tons and 14 billion US dollars, respectively [7,8]. The genus *Dioscorea* has over 600 species, of which 11 are economically significant [9].

In DR Congo, yam is a major staple of the rural community, whose major occupation is farming, and is a scarce food commodity in the major city markets due to insufficient

production capacity. Of the economically significant species, seven have been previously reported to play a major role in subsistence livelihoods: white guinea yam (*D. rotundata*), yellow guinea yam (*D. cayenensis*), water yam (*D. alata*), bitter yam (*D. dumetorum*), bush yam (*D. praehensilis*), wild yam (*D. burkilliana*), and aerial yam (*D. bulbifera*) [10–12]. Many of these species are being cultivated under wide agro-ecological zones, though with higher preference for *D. rotundata* and *D. alata* [10,13].

Despite the importance of yam in sustaining rural livelihoods, yam production is faced with lots of constraints, including, but not limited to, biotic (pests and diseases), tuber quality (oxidative browning, dry matter, and taste), and agronomic (yield) constraints [10,14,15]. Of the biotic constraints, pests (nematodes, beetles, etc.) and two major foliar diseases (yam anthracnose disease (YAD) and yam mosaic virus disease (YMV)) are the major contributors to production loss. These foliar diseases have been reported by the yam scientific community as major pathological problems to yam productivity and have resulted in the loss of many traditional cultivars (landraces) in many yam-producing countries [6,14,16]. In DR Congo, the extent of affliction has over the time been aggravated by the absence of improved (resistant/tolerant) varieties of yams and the inability of subsistence farmers to afford the cost of adequate control measures.

Agronomic attributes, such as yield potential, tuber shape, and tuber quality characteristics (e.g., tuber dry matter content and oxidative browning), in general, play a major role in the acceptance of yam varieties by farmers and consumers. Thus, these attributes have most often been regarded as farmers' and consumers' preference criteria, upon which research has been focused in recent decades [1,15]. As in every other yam-producing country, yam farmers in DR Congo also prefer yam varieties characterized by a combination of marketable yield, sweet tuber taste, zero to minimal tuber flesh oxidative browning, high tuber dry matter content, and tolerance to yam foliar diseases [10]. These attributes are mostly combined in improved yam genotypes following years of breeding efforts. Obtaining such varieties is an impossibility for most farmers as they depend on local varieties (landraces) for seasonal cultivation. Though ennoblement efforts by a few farmers has helped in identifying very few landraces with good agronomic and tuber quality attributes, the majority of the farmers still lack access to seeds of these landraces [10–12].

Yam production constraints in DRC have been aggravated by the lack of yam improvement programs to address challenges relating to yield improvement, resistance to foliar diseases, and post-harvest tuber quality improvement. In the absence of structured yam improvement programs to enhance the genetic potential of the existing traditional cultivars, as well as to develop new and improved yam cultivars, an alternative way to contribute to the improvement of farmers' productivity will be to assess the existing traditional cultivars for the criteria that are of the utmost importance to the farmers and consumers. This will allow the identification of landraces that combine good agronomic, tuber quality, and disease resistance attributes, and thus they can be recommended to the farmers for cultivation through the Ministry of Agriculture. Therefore, this study was carried out to (i) identify landraces (cultivated and semi-wild species) with superior performance for yam foliar disease resistance, agronomic, and tuber quality traits and (ii) estimate the components of variance and heritability for the traits considered in the study for selection purposes in future yam improvement programs.

## 2. Materials and Methods

### 2.1. Experimental Site, Planting Materials, Experimental Layout, and Planting

The study was carried out at two research places of the University of Kisangani (UNIKIS), namely Simi-Simi (longitude 0°33′05.9″ N, latitude 25°05′17.3″ E, altitude 396 m a.s.l, elevation 397 m a.s.l) and Akodali (longitude 0°35′46.4″ N, latitude 25°08′56.6″ E, altitude 419 m a.s.l, elevation 428 m a.s.l), Kisangani, DR Congo. The duration of the field evaluation lasted 11 months from April 2020. The evaluation sites are characterized by the dense humid forest vegetation with an irregularly distributed rainfall pattern throughout the year (3156 mm annual). The soil type in both locations is mostly oxisols (ferralsols ac-

cording to FAO classification) [17], and the mean temperature range is 21–35 °C minimum and maximum temperatures, respectively.

The planting materials consisted of a panel of 191 genotypes (188 landraces and three breeding lines) across six species of *Dioscorea* (Table 1). The morphotypes within each species vary in quantity in the following order: *D. rotundata* (108), *D. alata* (33), *D. dumetorum* (16), *D. praehensilis* (16), *D. cayenensis* (12), and *D. bulbifera* (6). The landraces were sourced from six territories (Kisangani, Isangi, Bumba, Lisala, Buta, and Bambesa), categorized within three provinces (Tshopo, Mongala, and Bas-Uele). The breeding lines included as standard checks were obtained from the yam breeding unit (yam improvement program) of the International Institute of Tropical Agriculture (IITA), Ibadan, Nigeria. These standard checks were of the *D. rotundata* (TDr9519177 and TDr8902665) and *D. alata* (TDa1100316) species with known pathological, agronomic, and tuber quality attributes potential.

**Table 1.** List of the panel of 191 yam accessions used for the trial evaluation.

| s/n | Accession Identity | Landrace Name | Territory | s/n | Accession Identity | Landrace Name | Territory | s/n | Accession Identity | Landrace Name | Territory |
|---|---|---|---|---|---|---|---|---|---|---|---|
| 1 | TDr21_001 | Libanza-1 | Bumba | 32 | TDr21_043 | Moindo-1 | Bumba | 63 | TDp21_052 | Ahala-28 | Bumba |
| 2 | TDr21_096 | Moenge-1 | Bumba | 33 | TDr21_067 | Ahala-12 | Bumba | 64 | TDr21_170 | Bozongo-4 | Bumba |
| 3 | TDr21_025 | Ahala-1 | Bumba | 34 | TDr21_027 | Ahala-13 | Bumba | 65 | TDp21_026 | Ahala-29 | Bumba |
| 4 | TDr21_141 | Libanza-2 | Bumba | 35 | TDr21_111 | Ahala-14 | Bumba | 66 | TDr21_112 | Bozongo-5 | Bumba |
| 5 | TDr21_010 | Ahala-2 | Bumba | 36 | TDr21_128 | Ahala-15 | Bumba | 67 | TDr21_074 | Ahala-30 | Bumba |
| 6 | TDr21_015 | Moenge-2 | Bumba | 37 | TDr21_016 | Ahala-16 | Bumba | 68 | TDr21_116 | Bozongo-6 | Bumba |
| 7 | TDr21_046 | Libanza-3 | Bumba | 38 | TDr21_044 | Libanza-11 | Bumba | 69 | TDr21_157 | Ahala-31 | Bumba |
| 8 | TDr21_158 | Ahala-3 | Bumba | 39 | TDr21_166 | Libanza-12 | Bumba | 70 | TDr21_187 | Ahala-32 | Bumba |
| 9 | TDr21_131 | Libanza-4 | Bumba | 40 | TDr21_097 | Moenge-5 | Bumba | 71 | TDr21_017 | Ahala-33 | Bumba |
| 10 | TDr21_177 | Ahala-4 | Bumba | 41 | TDc21_172 | Bwanzele-2 | Buta | 72 | TDr21_110 | Ahala-34 | Bumba |
| 11 | TDr21_179 | Moenge-3 | Bumba | 42 | TDr21_127 | Ahala-17 | Bumba | 73 | TDr21_020 | Libanza-15 | Bumba |
| 12 | TDr21_085 | Libanza-5 | Bumba | 43 | TDr21_012 | Ahala-18 | Bumba | 74 | TDr21_004 | Ahenge-1 | Bumba |
| 13 | TDc21_070 | Bwanzele-1 | Buta | 44 | TDr21_165 | Ahala-19 | Bumba | 75 | TDr21_167 | Ahala-35 | Bumba |
| 14 | TDr21_021 | Wasalaka | Bumba | 45 | TDr21_006 | Libanza-13 | Bumba | 76 | TDr21_164 | Libanza-16 | Bumba |
| 15 | TDr21_186 | Ahala-5 | Bumba | 46 | TDr21_175 | Moenge-6 | Bumba | 77 | TDr21_031 | Moenge-12 | Bumba |
| 16 | TDr21_033 | Bozongo-1 | Bumba | 47 | TDr21_109 | Ahala-20 | Bumba | 78 | TDr21_087 | Ahala-36 | Bumba |
| 17 | TDa21_084 | Ekolo-1 | Kisangani | 48 | TDr21_161 | Ahala-21 | Bumba | 79 | TDr21_013 | Ahenge-2 | Bumba |
| 18 | TDr21_047 | Bozongo-2 | Bumba | 49 | TDr21_093 | Moindo-2 | Bumba | 80 | TDr21_082 | Libanza-17 | Bumba |
| 19 | TDr21_181 | Moenge-4 | Bumba | 50 | TDr21_105 | Ahala-22 | Bumba | 81 | TDr21_089 | Ahala-37 | Bumba |
| 20 | TDr21_154 | Ahala-6 | Bumba | 51 | TDr21_101 | Moenge-7 | Bumba | 82 | TDr21_183 | Libanza-18 | Bumba |
| 21 | TDr21_108 | Libanza-6 | Bumba | 52 | TDr21_129 | Moenge-8 | Bumba | 83 | TDr21_191 | Ahala-38 | Bumba |
| 22 | TDc21_117 | Libanza-7 | Bumba | 53 | TDr21_106 | Moenge-9 | Bumba | 84 | TDr21_030 | Moenge-13 | Bumba |
| 23 | TDr21_045 | Ahala-7 | Bumba | 54 | TDc21_190 | Ngbongboto-1 | Buta | 85 | TDr21_155 | Ahala-39 | Bumba |
| 24 | TDr21_066 | Ahala-8 | Bumba | 55 | TDr21_024 | Ahala-23 | Bumba | 86 | TDr21_118 | Ahala-40 | Bumba |
| 25 | TDc21_059 | Libanza-8 | Bumba | 56 | TDr21_039 | Ahala-24 | Bumba | 87 | TDc21_091 | Libanza-19 | Bumba |
| 26 | TDr21_092 | Libanza-9 | Bumba | 57 | TDr21_113 | Ahala-25 | Bumba | 88 | TDr21_037 | Ahala-41 | Bumba |
| 27 | TDr21_119 | Ahala-9 | Bumba | 58 | TDr21_139 | Ahala-26 | Bumba | 89 | TDr21_057 | Libanza-20 | Bumba |
| 28 | TDr21_060 | Ahala-10 | Bumba | 59 | TDr21_140 | Libanza-14 | Bumba | 90 | TDr21_143 | Libanza-21 | Bumba |
| 29 | TDr21_007 | Bozongo-3 | Bumba | 60 | TDr21_171 | Ahala-27 | Bumba | 91 | TDr21_148 | Ahala-42 | Bumba |
| 30 | TDr21_083 | Libanza-10 | Bumba | 61 | TDr21_184 | Moenge-10 | Bumba | 92 | TDr21_142 | Engbo | Bumba |
| 31 | TDr21_162 | Ahala-11 | Bumba | 62 | TDr21_104 | Moenge-11 | Bumba | 93 | TDd21_174 | Biamajaune-1 | Kisangani |
| 94 | TDr21_153 | Ahala-43 | Bumba | 126 | TDd21_075 | Bilenge-2 | Kisangani | 158 | TDa21_080 | Ekolo-2 | Kisangani |
| 95 | TDr21_071 | Moenge-14 | Bumba | 127 | TDd21_124 | Bilenge-7 | Isangi | 159 | TDp21_063 | Lihoma | Isangi |
| 96 | TDr21_061 | Ahala-44 | Bumba | 128 | TDd21_069 | Bilenge-3 | Kisangani | 160 | TDa21_169 | Ekolo-3 | Kisangani |
| 97 | TDr21_051 | Moenge-15 | Bumba | 129 | TDd21_094 | Bilenge-4 | Kisangani | 161 | TDa21_098 | Ekolo-4 | Kisangani |
| 98 | TDr21_163 | Moenge-16 | Bumba | 130 | TDd21_103 | Gelenge | Kisangani | 162 | TDa21_073 | Ekolo-5 | Kisangani |
| 99 | TDr21_038 | Moenge-17 | Bumba | 131 | TDa21_133 | IFA_Kis-4 | Kisangani | 163 | TDa21_144 | Ekolo-6 | Kisangani |
| 100 | TDr21_134 | Libanza-22 | Bumba | 132 | TDa21_008 | Masua | Isangi | 164 | TDa21_009 | Ekolo-7 | Kisangani |
| 101 | TDr21_099 | Ahenge-3 | Bumba | 133 | TDp21_081 | Bainabaina | Isangi | 165 | TDa21_005 | Ekolo-8 | Kisangani |
| 102 | TDr21_041 | Ahala-45 | Bumba | 134 | TDp21_049 | Bosondi-3 | Isangi | 166 | TDa21_064 | Ekolo-9 | Kisangani |
| 103 | TDr21_115 | Ahulungenge-1 | Bumba | 135 | TDr21_088 | Ahala-48 | Bumba | 167 | TDa21_019 | Ekolo-10 | Kisangani |
| 104 | TDr21_102 | Ahulungenge-2 | Bumba | 136 | TDd21_136 | Bilenge-5 | Kisangani | 168 | TDa21_068 | Ekolo-11 | Kisangani |
| 105 | TDr21_062 | Ahenge-4 | Bumba | 137 | TDd21_011 | Bwanzele-3 | Buta | 169 | TDa21_032 | Ekolo-12 | Kisangani |
| 106 | TDr21_100 | Ahulungenge-3 | Bumba | 138 | TDd21_048 | Biamajaune-5 | Kisangani | 170 | TDa21_050 | Ekolo-16 | Kisangani |
| 107 | TDr21_053 | Ahulungenge-4 | Bumba | 139 | TDd21_029 | Biamajaune-6 | Kisangani | 171 | TDa21_014 | Ekolo-17 | Kisangani |
| 108 | TDr21_126 | Ahala-46 | Bumba | 140 | TDp21_028 | Bipaluma | Isangi | 172 | TDp21_159 | Bosondi-4 | Kisangani |
| 109 | TDr21_185 | Ahala-47 | Bumba | 141 | TDa21_079 | Biamawali-3 | Kisangani | 176 | TDp21_182 | Bwanzele-4 | Bambesa |
| 110 | TDr21_168 | Ahulungenge-5 | Bumba | 142 | TDa21_072 | Biamajaune-7 | Kisangani | 177 | TDp21_123 | Adia | Buta |
| 111 | TDr21_077 | Ahulungenge-6 | Bumba | 143 | TDd21_056 | Biamajaune-8 | Kisangani | 178 | TDp21_036 | Ambaga | Buta |
| 112 | TDc21_176 | Ahenge-5 | Bumba | 144 | TDd21_090 | Biamajaune-9 | Kisangani | 179 | TDp21_040 | Bwanzele-5 | Buta |
| 113 | TDr21_107 | IFA_Kis-1 | Kisangani | 145 | TDb21_022 | Litehu-1 | Kisangani | 180 | TDc21_147 | Bwanzele-6 | Buta |
| 114 | TDr21_137 | IFA_Kis-2 | Kisangani | 146 | TDb21_002 | Litehu-2 | Kisangani | 181 | TDa21_132 | Ekpego | Bambesa |
| 115 | TDr21_055 | IFA_Kis-3 | Kisangani | 147 | TDb21_086 | Litehu-3 | Kisangani | 182 | TDc21_035 | Ngbongboto-4 | Bambesa |

**Table 1.** *Cont.*

| s/n | Accession Identity | Landrace Name | Territory | s/n | Accession Identity | Landrace Name | Territory | s/n | Accession Identity | Landrace Name | Territory |
|---|---|---|---|---|---|---|---|---|---|---|---|
| 116 | TDc21_138 | Ngbongboto-2 | Buta | 148 | TDb21_130 | Litehu-4 | Kisangani | 183 | TDc21_135 | Manzaka | Buta |
| 117 | TDc21_188 | Ngbongboto-3 | Buta | 149 | TDa21_120 | Ilumbelumbe-1 | Kisangani | 184 | TDp21_121 | Bwanzele-7 | Buta |
| 118 | TDa21_149 | Biamawali-1 | Kisangani | 150 | TDa21_125 | Inene-1 | Kisangani | 185 | TDc21_018 | Bwanzele-8 | Buta |
| 119 | TDa21_189 | Biamawali-2 | Kisangani | 151 | TDa21_160 | Ilumbelumbe-2 | Kisangani | 186 | TDd21_146 | Avuadipudi | Lisala |
| 120 | TDa21_095 | Biamajaune-2 | Kisangani | 152 | TDa21_173 | Inene-2 | Kisangani | 187 | TDp21_058 | Mboloko | Lisala |
| 121 | TDa21_180 | Biamajaune-3 | Kisangani | 153 | TDr21_054 | TDr8902665 | IITA | 188 | TDp21_122 | Mokongo | Lisala |
| 122 | TDp21_065 | Bosondi-1 | Isangi | 154 | TDr21_151 | TDr9519177 | IITA | 189 | Tda21_150 | Maswe_1 | Lisala |
| 123 | TDp21_078 | Bosondi-2 | Isangi | 155 | TDa21_042 | TDa1100316 | IITA | 190 | Tda21_192 | Maswe_2 | Lisala |
| 124 | TDd21_114 | Biamajaune-4 | Kisangani | 156 | TDd21_156 | Bilenge-6 | Kisangani | 191 | TDb21_023 | Lihote | Lisala |
| 125 | TDd21_145 | Bilenge-1 | Kisangani | 157 | TDb21_076 | Liseleka | Isangi | | | | |

The experiment was conducted in a 12 by 16 lattice design with two replicates. The field layout was generated using "Agricolae" package in R [18]. Each replicate was comprised of 16 incomplete blocks with 12 experimental plots. In each replicate, the experimental unit was comprised of 5 m long ridges containing five plants at 1 m intra- and inter-row spacing. The planting was done with yam setts ranging between 150 to 200 g each, treated using a cocktail of fungicide (Mancozeb 7.5 g/liter of water) and insecticide (Cypermethrin 7.5 mL/liter of water). Following the sprouting of the planted setts, the plants were exposed to natural field infestation of yam mosaic virus and yam anthracnose disease, and no fertilizers were applied during the evaluation process. Weeding was done manually when necessary.

### 2.2. Data Collection

Data were collected on the traits of economic significance to farmers and consumers (Table 2).

**Table 2.** List of some traits assessed during the trial evaluation.

| S/N | Trait | Nature of the Trait | Collection Period | Collection Method |
|---|---|---|---|---|
| 1 | Plant vigor | Visual assessment of the vigor of the vine and leaves of the new plant in a plot | 4 MAP | Using a 1–3 scale where 1 = weak (75% of the plants or all the plants in a plot are small and have few leaves and thin vines), 2 = medium (intermediate or normal), and 3 = vigorous (75% of the plants or all the plants in a plot are robust, with thick vines and leaves very well developed or with abundant foliage) |
| 2 | Plant leaf density | Observation of variation in leaf mass or abundance on a mature plant and rating of density | 4 MAP | Using a 1–7 scale where 3 = low, 5 = intermediate, and 7 = high |
| 3 | Senescence class | Visual observation of the status of foliage senescence (leaf or vine yellowing) of plants in plot at 6 months and onward and rating of the maturity class (status) | 8 MAP | Using a 1–9 scale where 1 = very late (all the plants in a plot still show green foliage), 3 = late (75% of the plants in a plot still show green foliage, but up to 25% plants in a plot show leaves senescence), 5 = medium (50% of the plants still show green leaves and 50% showing senescence), 7 = early (75% of the plants in a plot show senescent foliage), and 9 = very early (all the plants in a plot are completely senesced). |
| 4 | Number of tubers per plot | The total quantity of the harvested tubers in a plot | At harvest | By count |
| 5 | Tuber size class | The average length of five tubers measured from the corm to the base in centimeters | At harvest | Using a 1–3 scale where 1 = small (less than 15 cm in length), 2 = medium (between 15 and 25 cm in length), and 3 = big/large (more than 25 cm in length) |
| 6 | Intensity of tuber flesh texture | The texture of tuber flesh after being cut | Post-harvest | Using a 1–3 scale where 1 = smooth, 2 = grainy, and 3 = very grainy |
| 7 | Intensity of tuber flesh oxidation | Degree of flesh surface color change or browning of cut yam tubers scored at different time intervals (0, 30, 60, 180 min) | Post-harvest | Using a 1–3 scale where 1 = no oxidization, 2 = slightly oxidizing, and 3 = highly oxidizing |

**Table 2.** *Cont.*

| S/N | Trait | Nature of the Trait | Collection Period | Collection Method |
|---|---|---|---|---|
| 8 | YMV severity | Visual assessment of the grade of reaction of the plant to the virus infection, varying from mottle, mosaics until total leaf deformation, recording of the severity as a proportion or percentage of plant surface affected | 2–6 MAS | Using a visual five ordinal scale (1–5 scale) where 1 = no visible symptoms; 2 = mosaic on few leaves, symptom recovery over time; 3 = mild symptoms on many leaves but no leaf distortion; 4 = severe mosaic on most leaves, leaf distortion; and 5 = severe mosaic (bleaching), severe leaf distortion and stunting |
| 9 | YADS severity | Visual assessment of anthracnose severity by observing the relative or absolute area of plant tissue affected by yam anthracnose disease and recording of the severity as a proportion or percentage of plant surface affected | 2–6 MAS | Using a visual 1–5 general scale where 1 = no visible symptoms of anthracnose disease, 2 = few anthracnose spots or symptoms on 1 to ~25% of the plant, 3 = anthracnose symptoms covering ~26 to ~50% of the plant, 4 = symptoms on >51% of the plant, and 5 = severe necrosis and death of the plant |

MAP = Month after planting; MAS = Month after sprouting.

The area under the disease progression curve (*AUDPC*), a valuable quantitative summary of disease severity for YMV and YAD over time, was estimated using the trapezoidal method [19]. This method discretizes the time variable and calculates the average disease severity between each pair of adjacent time points:

$$AUDPC = \sum_{i=1}^{N} \frac{(Yi + Yi + 1)}{2} (ti + 1 - ti) \tag{1}$$

where $N$ is the number of assessments made, $Y_i$ is the anthracnose or virus severity score on date $i$, and $t$ is the time in months between assessments $Y_i$ and $Y_{i+1}$.

Pathological reactions to yam mosaic virus (YMV) and to yam anthracnose disease (YAD) (severity scores) were recorded monthly from two to six months after sprout (Figure 1).

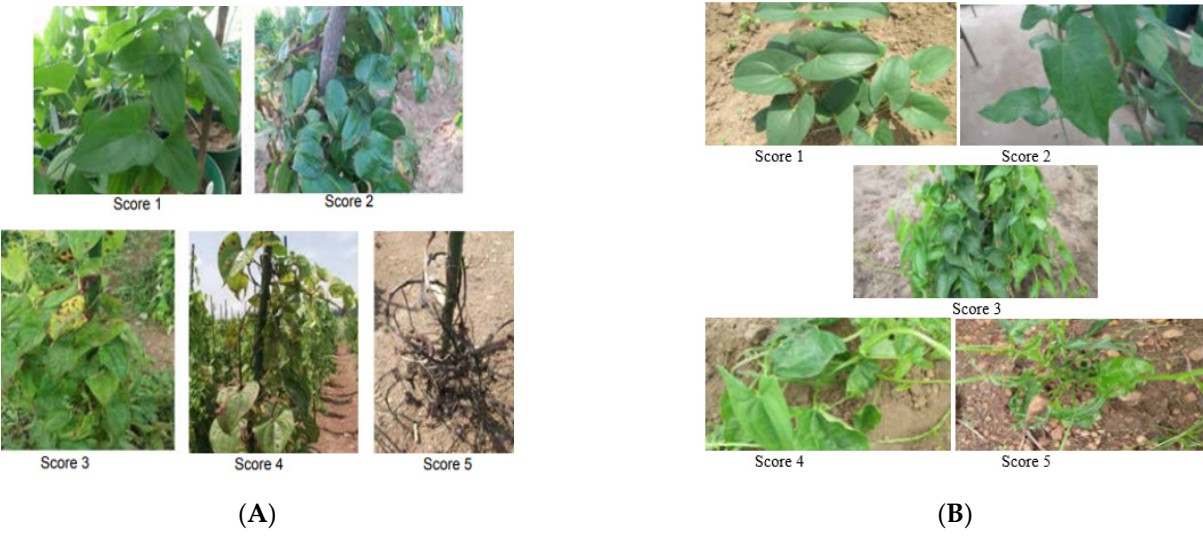

**(A)**                     **(B)**

**Figure 1.** Visual scale for yam anthracnose (**A**) and yam mosaic virus diseases (**B**) scoring (pictures from Asfaw, 2016 [20]).

The plant vigor and leaf density were assessed at two and three months after sprout emergence, respectively. Senescence class, a measure of maturity class was assessed at six months after sprout emergence. Parameters used for yield assessment at harvest included number of tubers harvested per plot, tuber size category, and fresh tuber and/or bulbil weight per plot. The intensity of tuber flesh oxidation, tuber flesh texture, and tuber dry matter content in percentage were collected post-harvest. All the traits were assessed using the recommendations of Asfaw, 2016 [20] and yam crop ontology: https://yambase.org/tools/onto/ (access on 25 February 2022).

Genotype fresh weight per plot was converted to the total tuber yield adjusted (*TTYA*) in tons per hectare using the formula below:

$$TTYA = \frac{TTWPx0}{PLS} \qquad (2)$$

where TTWP is the total tuber weight per plot, and *PLS* is the plot size.

Sett multiplication ration (*SMR*) was estimated as

$$SMR = \frac{\text{Weight of fresh tuber harvested}}{\text{Weight of sett planted}} \qquad (3)$$

The dry matter content (*DMC*) was determined by grating 200 g of fresh tuber flesh into a container and oven-drying it at 120 °C for 48 h, at which constant weight was observed. The percentage dry matter content was estimated as

$$\% \, DMC = \frac{\text{Dry tuber flesh weight}}{\text{Wet tuber flesh weight}} \times 100 \qquad (4)$$

*2.3. Statistical Analyses*

A mixed linear model was used to conduct analysis of variance (ANOVA) using the lmerTest package in R [21] following the alpha lattice model below:

$$Y_{ijkl} = \mu + Gen_i + Rep_j + Rep(Blk)_{j(k)} + Env_l + Gen \times Env_{(il)} + Error_{ijkl} \qquad (5)$$

where $Y_{ijk}$ is the phenotypic performance of accession for traits under consideration, $\mu$ is the average accession performance, $Gen_i$ is the effect of accession *i*, $Rep_j$ is the effect of replication *j*, $Rep(Blk)_{j(k)}$ is the block *k* effect nested in replication *j*, $Env_l$ is the effect of environment *l*, $Gen*Env_{(il)}$ is the effect of the accession *i* by environment *l* interaction, and $Error_{ijkl}$ is the residual effect.

For the analysis, accession (landrace) and environment were considered to be random effects while species was considered to be a fixed effect. Error ($\delta^2 e$), genotypic ($\delta^2 g$), and phenotypes ($\delta^2 p$) variances were calculated from expected mean squares (EMS) of ANOVA following Kresovich, 1990 [22].

Error variance;

$$\delta^2 e = \text{MSe}, \qquad (6)$$

Genotypic variance;

$$\delta^2 g = \frac{Msg - Msgl}{rl} \qquad (7)$$

Genotypic by environment interaction variance;

$$\delta^2 gl = \left( \frac{Msg - Msgl}{lr} \right) \qquad (8)$$

Phenotypic variance;

$$\delta^2 p = \delta^2 g + \left( \frac{\delta^2 e}{rl} \right) + \left( \frac{\delta^2 gl}{l} \right) \qquad (9)$$

where, *MSg* = mean square of genotype; *MSgl* = mean square due to accession by environmental interaction; MSe = error mean square (mean square of environment); *l* = number of locations/environment; *r* = number of replications.

Broad-sense heritability (*H*2), phenotypic coefficient of variance (*PCV*), and genotypic coefficient of variance (*GCV*) were calculated using the values derived from respective variance components. Broad-sense heritability (*H*2) was classified as low (<30%), medium (30–60%), and high (>60%), according to Johnson et al. [23]. Following Deshmukh et al. [24],

phenotypic and genotypic coefficients of variation greater than 20% were rated as high, between 10 and 20% were rated as medium, and lower than 10% were regarded as low.

$$H2 = \frac{\delta^2 g}{\delta^2 g + \frac{\delta^2 gl}{l} + \frac{\delta^2 e}{rl}} \times 100 \tag{10}$$

$$PCV = \left( \frac{\sqrt{\delta^2 p}}{\mu} \right) \times 100 \tag{11}$$

$$GCV = \left( \frac{\sqrt{\delta^2 g}}{\mu} \right) \times 100 \tag{12}$$

where $\delta^2 p$ = phenotypic variance, $\delta^2 g$ = genotypic variance, $\delta^2 gl$ = genotype by environment interaction variance; $\delta^2 e$: residual variance, r = number of replication; l = number of environment; μ: grand mean of the trait.

The relationship matrix, among the assessed traits, was constructed using Pearson's correlation coefficient and visualized using the ggpairs function in the GGally package [25]. Principal component analysis (PCA) was done using the PRCOMP function implemented in R [26] to identify the most discriminant traits with high contribution to the observed genotypic variation. Hierarchical cluster analysis was done based on the Ward.D2 method using the Gower dissimilarity matrix. The final hierarchical cluster was built and viewed using the dendextend package [27] and the circlize package [28] in R. The optimum number of clusters was identified using the NbClust package [29]. Path coefficient analysis was estimated and viewed using the lavaan function in the lavaan package [30]. In this model, tuber yield and tuber dry matter content were considered response variables against key agronomic and tuber quality traits as predictor variables. The path diagram was then constructed using the semPlot package [31] to depict the direct effect of these traits on tuber yield and dry matter content for suitability for indirect selection. Performance of landrace accession against check genotypes was assessed using Shukla's stability variance implemented in the VitSel application Version 1.0 [32].

## 3. Results

### 3.1. Variability in Agronomic and Tuber Quality Traits of Yam Landraces and Species

The analysis of variance (ANOVA) that shows the statistical difference for accessions and environment is presented in Table 3. Combined ANOVA revealed significant interaction effects of accession by environment at $p < 0.05$ for all the estimated parameters except for seed multiplication ratio, indicating environmental influence on the observed phenotypic expression of the landrace accessions for these traits. The interaction effect of species by environment was not significant for any parameter, suggesting that species performance was not environment dependent. Accession effect was significant at $p < 0.001$ for all the traits evaluated, indicating significant differences in the observed phenotypic performance of the accessions. Significant variation at $p < 0.05$ was observed for species effect in all the estimated parameters, indicating that the species performance differs for all the traits evaluated. Environment effect was significant for tuber dry matter content and yam anthracnose disease at $p < 0.01$, tuber oxidative browning at $p < 0.05$, and YMV severity at $p < 0.001$, indicating the existence of environmental differences with respect to these traits. Environment-specific analysis of variance revealed that both landrace and species effects were significant at $p < 0.001$ for all the studied traits in both environments.

**Table 3.** Combined and environment specific mean squares for agronomic and tuber quality traits in yam accessions.

| Source | DF | Yield (t/ha) | DMC | SMR | NTPP | TUBSZE | TUBOXI | FLSTXT | YMV | YAD | LFDEN | PLTVIG | SENSC |
|---|---|---|---|---|---|---|---|---|---|---|---|---|---|
| **Combined Environment** | | | | | | | | | | | | | |
| **Species** | 5 | 235.74 * | 215.50 *** | 162.74 *** | 20.86 ** | 1.93*** | 6.47 *** | 3.33 *** | 1.64 *** | 25.37 *** | 3.69 *** | 0.52 ** | 29.90 *** |
| **Env** | 1 | 69.04 | 131.51 ** | 134.55 | 22.15 | 0.12 | 1.46 * | 0.15 | 11.97 *** | 13.16 ** | 1.83 | 0.19 | 5.92 |
| **Accession** | 177 | 397.72 *** | 46.95 *** | 95.02 *** | 18.75 *** | 0.79 *** | 1.44 *** | 0.45 *** | 2.15 *** | 2.79 *** | 4.01 *** | 0.75 *** | 5.88 *** |
| **Species * Env** | 5 | 50.85 | 15.53 | 11.28 | 7.22 | 0.09 | 0.28 | 0.00 | 0.07 | 0.46 | 0.19 | 0.19 | 1.75 |
| **Env * Accession** | 177 | 137.51 ** | 13.86 * | 34.80 | 11.24 *** | 0.19 *** | 0.24 *** | 0.13 ** | 0.47 *** | 1.13 *** | 1.54 *** | 0.27 *** | 1.29 * |
| **Residual** | 335 | 91.71 | 11.01 | 31.02 | 6.54 | 0.12 | 0.15 | 0.09 | 0.31 | 0.53 | 0.77 | 0.13 | 0.99 |
| **CV%** | | 48.60 | 9.11 | 61.27 | 76.97 | 13.75 | 79.55 | 20.76 | 10.43 | 11.91 | 15.79 | 15.95 | 26.73 |
| **Mean** | | 20.49 | 35.35 | 9.20 | 3.35 | 2.58 | 0.48 | 1.41 | 5.23 | 6.30 | 5.60 | 2.31 | 3.75 |
| **Akodali environment** | | | | | | | | | | | | | |
| **Species** | 5 | 1205.61 *** | 891.25 *** | 596.08 *** | 52.03 *** | 4.84 *** | 16.16 *** | 18.66 *** | 13.58 *** | 32.20 *** | 20.82 *** | 3.11 *** | 86.32 *** |
| **Landrace** | 177 | 289.85 *** | 29.01 *** | 78.26 *** | 11.70 *** | 0.46 *** | 0.89 *** | 0.25 *** | 1.16 *** | 1.967 *** | 2.84 *** | 0.46 *** | 3.22 *** |
| **Residuals** | 152 | 100.06 | 9.00 | 32.51 | 3.52 | 0.12 | 0.19 | 0.10 | 0.30 | 0.46 | 0.63 | 0.11 | 1.20 |
| **CV%** | | 47.36 | 8.33 | 57.96 | 49.98 | 13.3 | 84.06 | 22.07 | 10.94 | 10.33 | 13.47 | 13.34 | 28.88 |
| **Mean** | | 21.36 | 36.09 | 9.97 | 3.70 | 2.61 | 0.52 | 1.40 | 5.06 | 6.56 | 5.88 | 2.44 | 3.83 |
| **Simi-Simi environment** | | | | | | | | | | | | | |
| **Species** | 5 | 878.19 *** | 655.61 *** | 413.15 *** | 114.25 *** | 7.11 *** | 14.43 *** | 18.53 *** | 9.68 *** | 36.83 *** | 20.15 *** | 4.23 *** | 99.18 *** |
| **Landrace** | 177 | 250.81 *** | 30.21 *** | 54.65 *** | 17.51 *** | 0.51 *** | 0.76 *** | 0.28 *** | 1.33 *** | 2.07 *** | 2.81 *** | 0.60 *** | 4.55 *** |
| **Residuals** | 152 | 83.40 | 12.76 | 26.30 | 9.54 | 0.14 | 0.13 | 0.09 | 0.30 | 0.59 | 0.89 | 0.15 | 0.80 |
| **CV%** | | 45.96 | 10.34 | 60.19 | 106.97 | 14.49 | 77.71 | 20.52 | 10.19 | 12.69 | 17.69 | 17.37 | 24.32 |
| **Mean** | | 19.62 | 34.60 | 8.43 | 2.97 | 2.56 | 0.45 | 1.41 | 5.41 | 6.03 | 5.32 | 2.18 | 3.68 |

DMC = Dry matter content; SMR = Sett multiplication ratio; NTPP = Number of tubers per plot; TUBSZE = Tuber size; TUBOXI = Intensity of tuber oxidation; FLSTXT = Tuber flesh texture; YMV = Yam mosaic virus disease; YAD = Yam anthracnose disease; LFDEN = Leaf density; PLTVIG = Plant vigor; SENSC = Senescence class; DF = Degree of freedom; CV = Coefficient of variation *, **, *** = significant at $p < 0.05$, 0.01, and 0.001 respectively.

Variation in the landrace species' mean performance in the combined analysis (Table 4) showed that *D. cayenensis* had the highest yield performance (28.49 t/ha) but was statistically similar to *D. alata* (25.72 t/ha) and significantly different from other species. *D. rotundata* had the highest dry matter content (37%), statistically similar to *D. cayenensis* (36.89%) and different from other species. *D. alata* had the highest set multiplication ratio (13.90), similar to that of *D. cayenensis* (11.73) but significantly different form other species. *D. alata* had the highest number of tubers per plot (4.55), similar to *D. bulbifera* but significantly different from other species. *D. cayenensis*, *D. rotundata*, *D. alata*, and *D. praehensilis* had significantly larger tuber size than the two other species. *D. alata* and *D. dumetorum* had the significantly minimal tuber flesh oxidative browning (−0.01 and −0.12, respectively), while *D. bulbifera* and *D. dumetorum* had significantly smoother flesh textures (0.90 and 0.94, respectively).

**Table 4.** Mean variations in agronomic and tuber quality traits of yam genotypes based on landrace species.

| Species | Yield (t/ha) | DMC | SMR | NTPP | TUBSZE | TUBOXI | FLSTXT | YMV | YAD | LFDEN | PLTVIG | SENSC |
|---|---|---|---|---|---|---|---|---|---|---|---|---|
| **Combined Environment** | | | | | | | | | | | | |
| TDa | 25.72 a | 35.17 bc | 13.90 a | 4.55 a | 2.65 a | −0.01 d | 2.41 a | 4.66 c | 7.72 a | 5.93 a | 2.22 b | 5.79 b |
| TDb | 16.53 b | 28.56 d | 1.62 c | 3.53 ab | 1.17 c | 0.71 bc | 0.90 d | 3.88 d | 6.54 c | 6.40 a | 2.76 a | 6.96 a |
| TDc | 28.49 a | 36.89 ab | 11.73 a | 3.04 b | 2.86 a | 1.07 b | 1.20 bc | 5.35 b | 5.95 cd | 6.03 a | 2.53 a | 2.12 e |
| TDd | 15.71 b | 25.84 e | 7.98 b | 2.71 b | 2.00 b | −0.12 d | 0.94 d | 4.49 c | 7.17 b | 5.66 ab | 2.59 a | 4.45 c |
| TDp | 20.83 b | 34.94 c | 8.23 b | 2.65 b | 2.65 a | 1.48 a | 1.37 b | 4.78 c | 5.59 d | 5.15 b | 2.26 b | 2.90 d |
| TDr | 18.77 b | 37.00 a | 8.15 b | 3.17 b | 2.69 a | 0.48 c | 1.20 c | 5.66 a | 5.84 d | 5.46 b | 2.26 b | 3.15 d |
| **Akodali environment** | | | | | | | | | | | | |
| TDa | 29.09 a | 35.65 b | 15.77 a | 4.49 a | 2.65 a | 0.06 c | 2.37 a | 4.58 b | 7.79 a | 6.28 a | 2.41 ab | 5.80 b |
| TDb | 17.83 bc | 28.61 c | 1.48 c | 3.59 ab | 1.26 c | 0.61 b | 0.85 d | 3.78 c | 6.92 bc | 6.45 a | 2.78 a | 7.07 a |
| TDc | 29.34 a | 38.28 a | 12.16 ab | 3.24 ab | 2.82 a | 1.32 a | 1.22 c | 5.20 a | 6.30 cd | 6.36 a | 2.57 ab | 2.67 d |
| TDd | 16.74 c | 25.91 c | 8.47 b | 2.67 b | 1.99 b | −0.16 c | 0.98cd | 4.48 b | 7.72 ab | 5.82 ab | 2.66 a | 4.55 c |
| TDp | 21.53 b | 34.90 a | 8.76 b | 2.95 b | 2.63 a | 1.51 a | 1.47 b | 4.65 b | 5.92 d | 5.21 b | 2.33 b | 3.32 d |
| TDr | 18.90 bc | 37.97 a | 8.77 b | 3.76 ab | 2.74 a | 0.51 b | 1.18 c | 5.41 a | 6.07 cd | 5.76 ab | 2.40 ab | 3.13 d |
| **Simi-Simi environment** | | | | | | | | | | | | |
| TDa | 22.36 ab | 34.69 a | 12.03 a | 4.61 a | 2.66 a | −0.08 d | 2.44 a | 4.73 b | 7.64 a | 5.56 ab | 2.03 c | 5.77 b |
| TDb | 15.24 bc | 28.51 b | 1.75 c | 3.46 ab | 1.07 c | 0.80 bc | 0.95 bc | 3.98 c | 6.15 bc | 6.35 a | 2.75 a | 6.85 a |
| TDc | 27.63 a | 35.51 a | 11.3 a | 2.84 ab | 2.89 a | 0.81 b | 1.17 b | 5.50 a | 5.59 c | 5.70 ab | 2.48 ab | 1.57 e |
| TDd | 14.68 c | 25.76 b | 7.49 b | 2.75 ab | 2.02 b | −0.09 d | 0.88 c | 4.50 bc | 6.62 b | 5.49 ab | 2.52 ab | 4.35 c |
| TDp | 20.12 abc | 34.98 a | 7.69 b | 2.35 b | 2.68 a | 1.44 a | 1.27 b | 4.90 b | 5.26 c | 5.07 b | 2.20 bc | 2.48 d |
| TDr | 18.64 bc | 36.02 a | 7.52 b | 2.58 b | 2.64 a | 0.46 c | 1.21 b | 5.91 a | 5.60 c | 5.16 b | 2.11 c | 3.18 d |

DMC = Dry matter content; SMR = Sett multiplication ratio; NTPP = Number of tubers per plot; TUBSZE = Tuber size; TUBOXI = Intensity of tuber oxidation; FLSTXT = Tuber flesh texture; YMV = Yam mosaic virus disease; YAD = Yam anthracnose disease; LFDEN = Leaf density; PLTVIG = Plant vigor; SENSC = Senescence class. TDa = Tropical *Dioscorea alata*; TDb = Tropical *Dioscorea bulbifera*; TDc = Tropical *Dioscorea cayenensis*; TDd = Tropical *Dioscorea dumetorum*; TDp = Tropical *Dioscorea praehensilis*; TDr = Tropical *Dioscorea rotundata*. The letters a, b, c, d & e represent the LSD level of significance.

Response to pathological disease revealed that *D. bulbifera* had the highest tolerance to YMV severity (AUDPC = 3.88), while *D. rotundata* had the least tolerance (AUPDC = 5.66). However, *D. praehensilis* (AUDPC = 5.59) and *D. rotundata* (AUDPC = 5.84) had the highest tolerance to YAD severity, while *D. alata* had the least tolerance (AUDPC = 7.72). *D. bulbifera*, *D. cayenensis*, and *D. alata* had significantly higher leaf density (6.40, 6.30, and 5.93, respectively), while *D. bulbifera*, *D. dumetorum*, and *D. cayenensis* had significantly better plant vigor (2.76, 2.59, and 2.53, respectively). *D. bulbifera* had significantly higher senescence class (6.96 = early maturing), while *D. cayenensis* had the lowest rating (2.12 = very late maturing) (Table 4).

### 3.2. Genetic Variability and Broad-Sense Heritability of Agronomic and Tuber Quality Traits Yam Accessions

Genotypic and phenotypic variance components, genotypic and phenotypic coefficients of variation, and broad-sense heritability of agronomic and tuber quality traits in yam accessions are presented in Table 5. Genotypic coefficients of variation (GCV) ranged from a moderate classification of 12.96% for tuber dry matter content to high classification 52.16% for tuber flesh oxidative browning. A similar result was observed for phenotypic coefficients of variation (PCV) which ranged from a moderate classification of 14% for tuber dry matter content to a high classification of 68.49% for the number of tubers per plant. Broad-sense heritability ($H^2$) varied between 46.97% (moderate) and 91.40% (high). High $H^2$ (>60%) was observed in all the estimated parameters except for number of tubers per plot, where moderate $H^2$ was observed.

**Table 5.** Genetic variance, coefficient of variation, and broad-sense heritability in yam landrace accessions.

| Traits | Genetic Parameters | | | | |
| --- | --- | --- | --- | --- | --- |
| | $\delta^2 g$ | $\delta^2 p$ | GCV (%) | PCV (%) | H$^2$ (%) |
| Yield (t/ha) | 80.20 | 114.94 | 43.71 | 52.32 | 69.78 |
| DMC | 21.00 | 24.48 | 12.96 | 14.00 | 85.77 |
| SMR | 22.43 | 31.03 | 51.50 | 60.57 | 72.30 |
| NTPP | 2.45 | 5.22 | 46.92 | 68.49 | 46.97 |
| TUBSZE | 0.25 | 0.30 | 19.34 | 21.19 | 84.12 |
| TUBOXI | 0.53 | 0.59 | 52.16 | 54.86 | 89.45 |
| FLSTXT | 0.35 | 0.39 | 42.08 | 44.42 | 91.40 |
| YMV | 0.60 | 0.72 | 14.80 | 16.22 | 83.93 |
| YAD | 0.86 | 1.15 | 14.73 | 17.03 | 74.34 |
| LFDEN | 0.81 | 1.19 | 16.07 | 19.48 | 67.85 |
| PLTVIG | 0.16 | 0.23 | 17.31 | 20.75 | 69.00 |
| SENSC | 2.76 | 3.10 | 44.25 | 46.90 | 88.97 |

$\delta^2 g$ = Genotypic variance; $\delta^2 p$ = Phenotypic variance; GCV = Genotypic coefficient of variation; PCV = Phenotypic coefficient of variation; H$^2$ = Broad-sense heritability; DMC = Dry matter content; SMR = Sett multiplication ratio; NTPP = Number of tubers per plot; TUBSZE = Tuber size; TUBOXI = Intensity of tuber oxidation; FLSTXT = Tuber flesh texture; YMV= Yam mosaic virus disease; YAD = Yam anthracnose disease; LFDEN = Leaf density; PLTVIG = Plant vigor; SENSC = Senescence class.

### 3.3. Principal Component Analysis of Agronomic and Tuber Quality Traits

The principal component analysis that was used to identify the most discriminant traits with high contributions to the observed genotypic variation is presented in Table 6. The first four principal components (PC), with Eigen values greater than one, accounted for 66.21% of the genetic variation in the study. The first PC accounted for 28.49% of variance, with major contributions from tuber yield, seed multiplication ratio, tuber size, leaf density, and plant vigor. The second PC accounted for 16.39%, with major contributions from number of tubers per plot, tuber size, tuber flesh texture, YAD severity, and senescence class. The third PC accounted for 12.68%, with major contributions from tuber dry matter content, tuber size, tuber flesh oxidative browning, YMV severity, and plant vigor. The fourth PC accounted for 8.65%, with major contributions from tuber dry matter content, tuber size, tuber flesh oxidative browning, tuber flesh texture, and YMV severity (Table 6).

**Table 6.** Principal component analysis and contributions of agronomic and tuber quality traits to the genetic variability.

| Trait | PC1 | PC2 | PC3 | PC4 |
| --- | --- | --- | --- | --- |
| Yield (t/ha) | **0.457** | −0.182 | 0.003 | −0.120 |
| DMC | 0.011 | −0.232 | **−0.437** | **0.252** |
| SMR | **0.455** | −0.076 | −0.123 | −0.158 |
| NTPP | 0.166 | **0.287** | −0.110 | 0.134 |
| TUBSZE | **0.316** | −0.264 | **−0.322** | **0.227** |
| TUBOXI | 0.076 | −0.162 | **0.327** | **0.762** |
| FLSTXT | 0.196 | **0.289** | **−0.391** | **0.345** |
| YMV | −0.117 | −0.221 | **−0.506** | **−0.284** |
| YAD | 0.220 | **0.472** | −0.123 | −0.016 |
| LFDEN | **0.452** | 0.019 | 0.142 | −0.128 |
| PLTVIG | **0.382** | −0.065 | **0.352** | −0.177 |
| SENSC | −0.003 | **0.608** | −0.052 | 0.009 |
| Eigen value | 1.849 | 1.402 | 1.234 | 1.019 |
| Variance (%) | 28.490 | 16.390 | 12.680 | 8.654 |
| Cumulative (%) | 28.490 | 44.880 | 57.560 | 66.213 |

DMC = Dry matter content; SMR = Sett multiplication ratio; NTPP = Number of tubers per plot; TUBSZE = Tuber size; TUBOXI = Intensity of tuber oxidation; FLSTXT = Tuber flesh texture; YMV = Yam mosaic virus disease; YAD = Yam anthracnose disease; LFDEN = Leaf density; PLTVIG = Plant vigor; SENSC = Senescence class. PC1 to PC12 indicate Principal Components.

### 3.4. Relationships among Agronomic and Tuber Quality Traits in Yam Landraces

The relationship among evaluated yam parameters is presented in Figure 2. A significant positive relationship was observed between tuber yield and sett multiplication ratio (r = 0.79), tuber size (r = 0.49), plant leaf density (r = 0.63) at $p < 0.001$, tuber flesh texture (r = 0.22) at $p < 0.01$, and number of tubers per plot (r = 0.16) at $p < 0.05$. A significant negative relationship was not observed. Dry matter content showed a significant positive relationship with tuber size (r = 0.28) and YMV severity (r = 0.41) at $p < 0.001$, while a negative relationship was observed for number of tubers per plot (r = −0.15, $p < 0.05$), YAD severity (r = −0.20), and leaf density (r = −0.23) at $p < 0.01$. A significant positive relationship indicates similar direction in trait performance, while a significant negative relationship indicates opposite direction in traits expression.

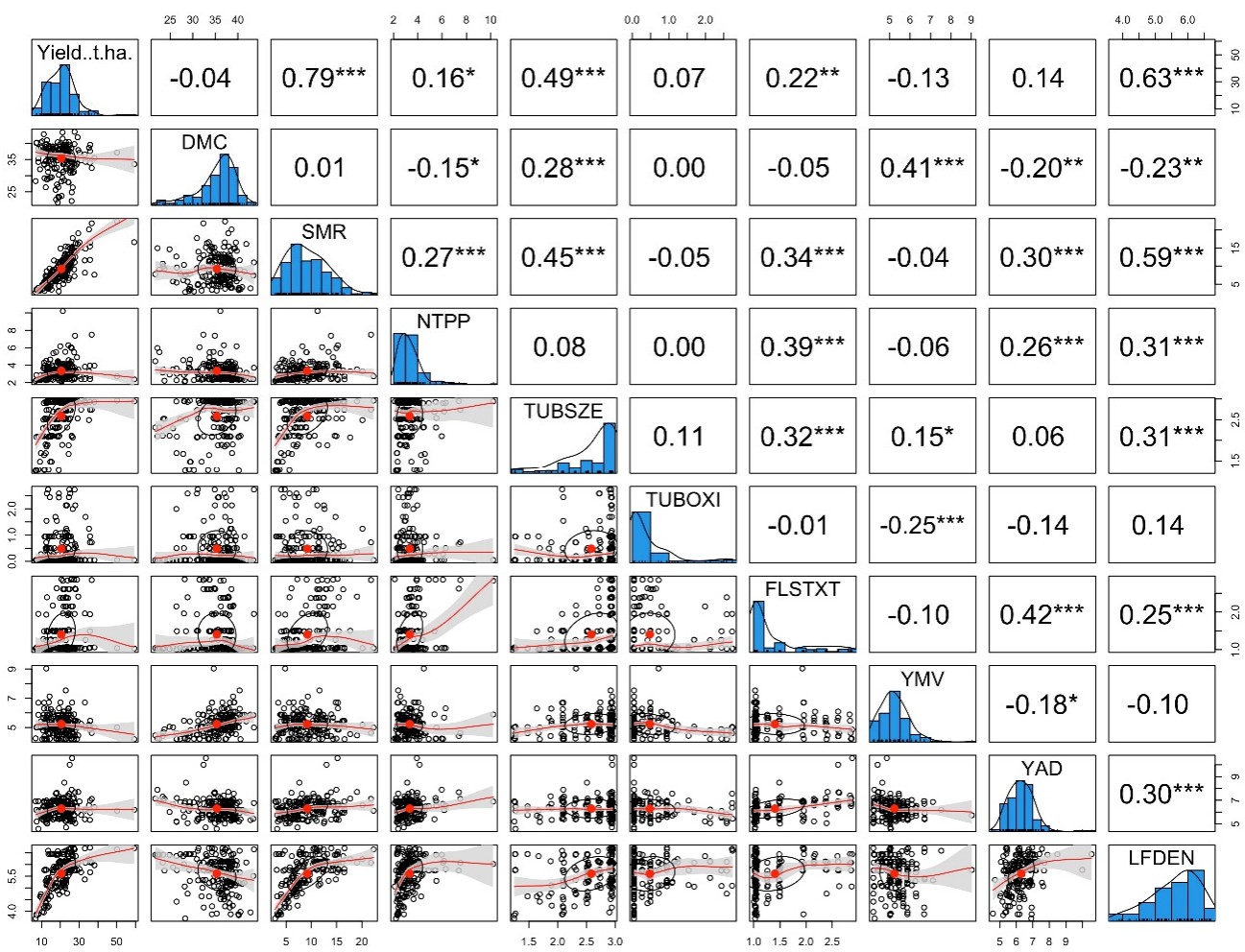

**Figure 2.** Correlation coefficients among agronomic and tuber quality traits. DMC = Dry matter content; SMR = Sett multiplication ratio; NTPP = Number of tubers per plot; TUBSZE = Tuber size; TUBOXI = Intensity of tuber oxidation; FLSTXT = Tuber flesh texture; YMV = Yam mosaic virus disease; YAD = Yam anthracnose disease; LFDEN = Leaf density; PLTVIG = Plant vigor; SENSC = Senescence class. *, **, *** = significant at $p < 0.05$, 0.01, and 0.001 respectively.

### 3.5. Yam Clustering Based on Hierarchical Clustering

Hierarchical clustering employed for the grouping of yam accessions based on the evaluated agronomic and tuber quality characters produced four clusters (Figure 3). Cluster one consisted of accessions of *D. alata* (30), characterized by high tuber yield, a high number of tubers per plot, high leaf density, large tuber size, low tuber flesh oxidative browning, very grainy tuber flesh texture, high susceptibility to YAD severity, and medium senescence

class. Cluster two had the largest cluster membership, which consisted of accession of *D. rotundata* (42), *D. praehensilis* (13), *D. cayenensis* (11), *D. alata* (1), and *D. dumetorum* (1), characterized by high yield, high dry matter content, large tuber size, high leaf density, high plant vigor, smooth tuber flesh texture, and moderate tuber oxidative browning. Cluster three had the minimum cluster members and consisted of accessions of *D. dumetorum* (13), *D. bulbifera* (6), and *D. alata* (3), characterized by a high number of tubers per plot, high leaf density, high plant vigor, low tuber flesh oxidative browning, smooth tuber flesh texture, early senescence class, and moderate tolerance to YMV severity but susceptibility to YAD severity. Cluster four consisted of accessions of *D. rotundata* (60) and *D. cayenensis* (1), characterized by high dry matter content, low tuber flesh oxidative browning, smooth tuber flesh textures, and susceptibility to YMV severity with moderate tolerance of YAD severity (Table 7).

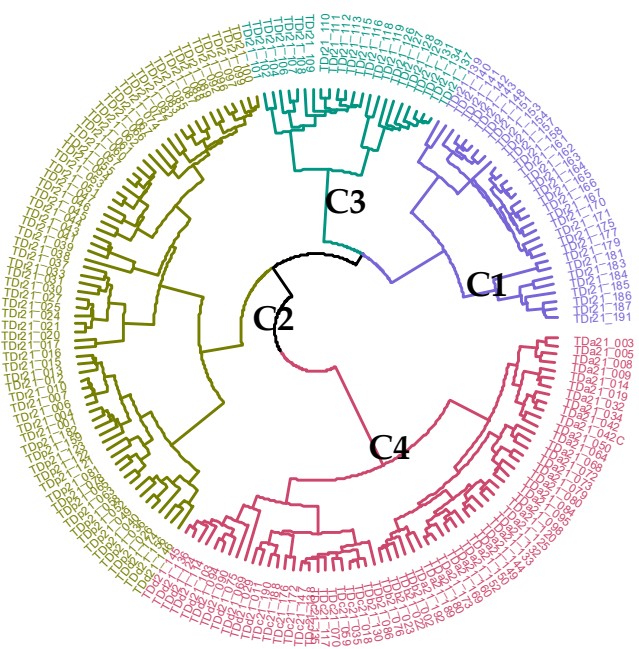

**Figure 3.** Hierarchical clustering showing grouping patterns of yam landrace accessions into four clusters using twelve key traits covering agronomic and tuber quality based on the Gower dissimilarity matrix. C1, Cluster one (blue); C2,; Cluster two (yellow); C3, Cluster three (green); C4, Cluster four (red). D. praehensilis (2).

**Table 7.** Description of clusters of yam landraces.

| Trait | Cluster 1 (30) | Cluster 2 (68) | Cluster 3 (22) | Cluster 4 (63) | F-Value |
|---|---|---|---|---|---|
| Tuber yield (t/ha) | **23.90** [a] | **24.79** [a] | 18.41 [b] | 15.01 [c] | 33.31 *** |
| Dry matter content (%) | 35.57 [b] | 36.32 [ab] | 27.26 [c] | **37.02** [a] | 62.88 *** |
| Sett multiplication ratio | **12.56** [a] | 10.85 [b] | 7.98 [c] | 6.25 [d] | 36.61 *** |
| Number of tuber per plot | **3.99** [a] | 3.35 [b] | 3.55 [ab] | 2.95 [c] | 7.68 *** |
| Tuber size | **2.87** [a] | **2.82** [a] | 1.91 [c] | 2.43 [b] | 50.11 *** |
| Tuber oxidative browning | 0.26 [b] | 0.88 [a] | **0.31** [b] | **0.22** [b] | 14.72 *** |
| Tuber flesh texture | 2.49 [a] | **1.23** [b] | **1.12** [b] | **1.18** [b] | 154.52 *** |
| Yam mosaic virus disease | 5.01 [b] | 5.27 [b] | **4.50** [c] | 5.56 [a] | 16.55 *** |
| Yam anthracnose disease | 7.12 [a] | **6.06** [b] | 6.83 [a] | **5.98** [b] | 28.63 *** |
| Leaf density | **6.01** [a] | 5.91 [a] | 6.03 [a] | 4.93 [b] | 47.36 *** |
| Plant vigor | 2.36 [b] | **2.49** [a] | **2.54** [a] | 2.02 [c] | 48.68 *** |
| Senescence class | **5.08** [a] | 2.85 [c] | **5.72** [a] | 3.42 [b] | 48.84 *** |

Significance level: "$p < 0.001$" = ***. Means followed by the same superscripts are not significantly different using the least significant difference (LSD) test at a 5% $p$-value threshold. The bold values indicate significant traits at each cluster. The letters a, b & c represent the LSD level of significance.

### 3.6. Path Analysis among Assessed Traits of Dioscorea Species

The path analysis done to depict the direct effect of agronomic traits on tuber yield and dry matter content for suitability for indirect selection is presented in Figure 4. The path analysis began with structural equation modelling where tuber yield and dry matter content were considered response variables against correlated agronomic and tuber quality parameters. The model resulted in excellent fit. The chi-square test of the model fit was not significant ($\chi^2$ (4) = 2.455, $p$ = 0.653). Overall, fit indices were in good range (RMSEA = 0.00 [0.00, 0.09], $p$ = 0.81; CFI = 1.00; SRMR = 0.01). Most of the direct effects in the model were significant.

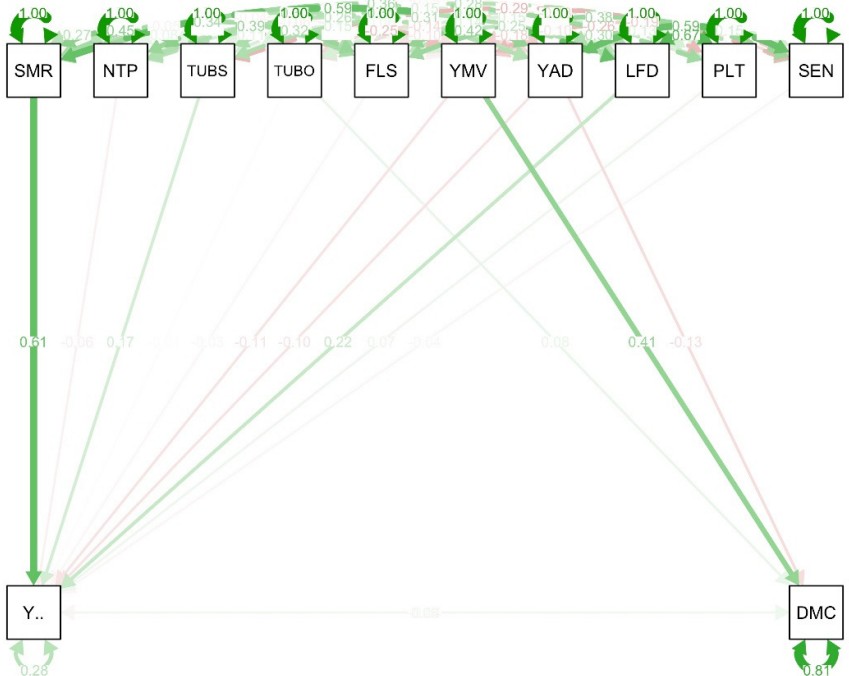

**Figure 4.** Path coefficient analysis between response and independent yam variables. Y.. = Tuber yield; DMC = Dry matter content; SMR = Sett multiplication ratio; NTP = Number of tubers per plot; TUB = Tuber size; TUBO = Tuber oxidation; FLS = Tuber flesh texture; YMV = Yam mosaic virus disease; YAD = Yam anthracnose disease; LFD = Leaf density; PLT = Plant vigor; SEN = Senescence class. Red indicates direct negative impact, and green indicates direct positive impact.

Setts multiplication ratio significantly predicted tuber yield (b = 1.12, SE = 0.10, $p$ < 0.001) such that a unit increase in setts multiplication ratio was associated with a 1.12-unit increase in tuber yield. Tuber size significantly predicted tuber yield (b = 2.64, SE = 0.83, $p$ < 0.001) such that a one-unit increase in tuber size was associated with a 2.64-unit increase in tuber yield. YMV severity significantly predicted tuber yield (b = −1.21, SE = 0.45, $p$ < 0.01) such that a one-unit increase in YMV severity was associated with a 1.21-unit decrease in tuber yield. Leaf density significantly predicted tuber yield (b = 2.08, SE = 0.65, $p$ < 0.001) such that a one-unit increase in leaf density was associated with a 2.08-unit increase in tuber yield (Figure 3).

Tuber size significantly predicted tuber dry matter content (b = 2.21, SE = 0.60, $p$ < 0.001) such that a one-unit increase in tuber size was associated with a 2.21-unit increase in tuber dry matter content. YMV severity significantly predicted tuber dry matter content (b = 1.54, SE = 0.39, $p$ < 0.001) such that a one-unit increase in YMV severity was associated with a 1.54-unit increase in tuber dry matter content. Plant vigor significantly predicted tuber dry matter content (b = −3.70, SE = 0.84, $p$ < 0.001) such that a one-unit increase in plant vigor was associated with a 3.70-unit decrease in tuber dry matter. Senescence class significantly predicted tuber dry matter content (b = −0.60, SE = 0.17, $p$ < 0.001) such that a one-unit increase in senescence class was associated with a 0.60-unit decrease in tuber dry matter content.

### 3.7. Performance of Landrace Accession against Standard Check Genotypes

Landraces performances for traits of interest were compared to that of the average performance of the three checks used in the study. Tuber yield and tuber dry matter were set as traits of higher values, and AUDPC estimates for YMV and YADS were set as traits of lower values, while tuber flesh oxidation was set as trait of values within range. The performance of landrace accession against the standard check genotypes revealed that 51 landrace accessions with Shukla's stability variances varying from 0 to 389 had better performance than the average performance of the three checks included in the study (Table S1). Of the 51 landraces accessions, only 20 accessions were observed to have stable performance (Shukla's stability variance of less than 5) with respect to the parameters under assessment. The 20 accessions were observed to be accessions of *D. alata* (TDa21_169; TDa21_080; TDa21_73; TDa21_152; TDa21_050; TDa21_005; TDa21_034), *D. cayenensis* (TDc21_059; TDc21_190), and *D. rotundata* (TDr21_162; TDr21_089; TDr21_142; TDr21_037; TDr21_153; TDr21_167; TDr21_154; TDr21_134; TDr21_131; TDr21_163; TDr21_099) (Table 8).

**Table 8.** List of the stable twenty landrace accessions with better performance over checks mean for farmers' and consumers' preferred traits.

| S/N | Genotype | Yield | DMC | YMV | YAD | TUBOXI | Stability | Rank |
|-----|----------|-------|------|------|------|--------|-----------|------|
| 1 | TDa21_169 | 25.88 | 38.43 | 5.68 | 6.89 | 0.80 | 0.00 | 0 |
| 2 | TDr21_162 | 21.19 | 43.40 | 5.85 | 5.44 | 0.05 | 0.00 | 0 |
| 3 | TDc21_059 | 19.30 | 38.00 | 6.32 | 6.11 | 0.06 | 0.15 | 7 |
| 4 | TDr21_089 | 21.25 | 41.43 | 5.87 | 6.08 | 0.95 | 0.17 | 11 |
| 5 | TDr21_142 | 17.74 | 38.54 | 5.02 | 6.12 | 0.05 | 0.18 | 13 |
| 6 | TDr21_037 | 21.58 | 36.58 | 5.86 | 6.70 | 0.05 | 0.35 | 19 |
| 7 | TDa21_080 | 21.35 | 36.10 | 5.89 | 6.89 | 0.05 | 0.46 | 22 |
| 8 | TDr21_153 | 18.61 | 40.32 | 5.86 | 6.87 | 0.05 | 0.46 | 24 |
| 9 | TDr21_167 | 24.39 | 37.23 | 5.84 | 6.12 | 0.95 | 0.48 | 25 |
| 10 | TDr21_154 | 25.52 | 36.30 | 5.87 | 5.96 | 0.27 | 1.18 | 46 |
| 11 | TDc21_190 | 33.08 | 36.54 | 6.71 | 5.40 | 0.05 | 1.59 | 52 |
| 12 | TDr21_134 | 20.11 | 39.09 | 6.71 | 5.47 | 0.05 | 1.60 | 54 |
| 13 | TDa21_073 | 23.35 | 38.21 | 5.47 | 6.56 | 0.05 | 1.72 | 57 |
| 14 | TDa21_152 | 23.84 | 38.05 | 5.05 | 6.90 | 0.28 | 2.06 | 64 |
| 15 | TDr21_131 | 25.25 | 39.39 | 5.03 | 5.38 | 0.05 | 2.18 | 67 |
| 16 | TDa21_050 | 19.85 | 36.74 | 5.47 | 6.71 | 0.05 | 2.32 | 68 |
| 17 | TDa21_005 | 21.75 | 38.87 | 5.26 | 7.12 | 0.13 | 2.98 | 75 |
| 18 | TDr21_163 | 33.02 | 36.54 | 5.04 | 6.22 | 0.05 | 3.05 | 77 |
| 19 | TDa21_034 | 24.09 | 37.78 | 5.26 | 6.90 | 0.20 | 3.16 | 78 |
| 20 | TDr21_099 | 35.81 | 37.57 | 5.04 | 6.96 | 0.95 | 3.46 | 80 |
|  | Checks mean | 17.40 | 35.42 | 6.72 | 7.21 |  |  |  |

DMC = Dry matter content; TUBOXI = Intensity of tuber oxidation; YMV = Yam mosaic virus disease; YAD = Yam anthracnose disease; Stability= Shukla's stability variance.

## 4. Discussion

### 4.1. Variability in Agronomic and Tuber Quality Traits of Dioscorea Species as Identifiers of Gene Reservoirs for Yam Genetic Improvement in DR Congo

Yam production in DR Congo is challenged by numerous constraints, including, but not limited to, low yield, poor tuber quality characteristics, and pathological diseases, which have been the major focus of modern breeding programs in countries where they exist. The identification of the genetic potential and gene reservoir for genetic improvement from the existing genetic pool of landraces for high yield potential, good tuber quality attributes, and resistance/tolerance to pathological diseases could offer a potential hope for consideration for yam improvement. The study revealed varying degrees of potential of the landrace species for farmers' and consumers' preferred traits (high yield, high dry matter content, resistance to pathological diseases, and non to low tuber oxidative browning). Accessions of *D. cayenensis* and *D. alata* had the highest yield potential among all the species considered. These species are popular for their high plant vigor and leaf density, which enhance the

yield potentials, hence the reason for the wide distribution of *D. alata* worldwide [1,33]. In addition, *D. cayenensis* requires a longer cycle, which is not a desired trait to many yam cultivators but allows the advantage of more assimilates production and translocation into the tubers compared to most cultivated species. Accessions of *D. rotundata* had the highest dry matter content and were probably the reason for the preference for consumption and industrial potential in many yam-producing communities [34,35]. Accessions of *D. alata* and *D. dumetorum* had very low tuber flesh oxidative browning properties compared to other species. This trait has been reported as a determinant in yam cultivar acceptability in many studies [15].

For resistance to pathological diseases, accessions of *D. bulbifera* had the best genetic tolerance to YMV disease; however, this species is not known for regular cultivation, as it is regarded as forest/wild species [36,37]. Of the cultivated species, accessions of *D. dumetorum*, *D. alata*, and *D. praehensilis* had better tolerance than the popular preference species for consumption (*D. rotundata*) and, as such, can be considered in breeding programs for the improvement of *D. rotundata*, particularly *D. praehensilis* due to their similar genome information [38]. This corroborates the findings of Adewumi et al. [39], who observed better tolerance of *D. praehensilis* genotypes over that of *D. rotundata* for YMV severity. Accessions of *D. praehensilis* and *D. rotundata* had the best genetic tolerance to YAD severity among the considered species while *D. alata* had the least tolerance. *D. alata* is very susceptible to YAD severity [16,40,41].

### 4.2. Genetic Parameters and Broad-Sense Heritability of Evaluated Traits

The high GCV and PCV (>20%) observed in some of the evaluated traits, such as tuber yield, seed multiplication ratio, number of tubers per plot, tuber flesh oxidative browning, tuber flesh texture, and senescence class indicates potentials for high selection intensity. This is essential, as it will facilitate the selection of accessions with superior performance in yam breeding programs. High GCV and PCV recorded for tuber yield in this study were in agreement with the findings of Padhan and Panda [42] conducted on advanced breeding populations of white yam. High $H^2$ (>60%) recorded in this study for all traits except for number of tubers per plot indicates a high correspondence between phenotypic and genotypic variance and, therefore, high response to selection. Many studies have also obtained similar findings for some of the observed parameters. Agre et al. [1] observed high $H^2$ estimates for tuber yield per plant and YMV in *D. rotundata*. Bhattacharjee et al. [16] also reported high broad-sense heritability for YAD in *D. alata*.

### 4.3. Correlation Coefficients, Principal Components, and Hierarchical Clusters among Assessed Traits of Landrace Accessions

Landraces accession with high leaf density, large tuber size, high seed multiplication ratio, grainy flesh texture, and high number of tubers per plot could be considered in breeding for improved yield following their observed relationship in the study. In consideration for improved tuber dry matter content, landrace accessions with a reduced number of tubers per plot, larger tuber size, tolerance to YAD severity, and reduced leaf density could be considered following the relationship observed in our study. Agre et al. [43] similarly observed a positive relationship between tuber yield and yield components, such as tuber size and number of tubers per plot in a panel of water yam.

The traits that best discriminated the landrace accession in this study were those which resolved on PC1 with major contribution. These traits, including tuber yield, tuber size, leaf density, plant vigor, YMV severity, YAD severity, and flesh texture could be utilized in evaluating genetic diversity among similar species of yam. Agre et al. [43,44] and Siadjeu, et al. [45] have previously reported the significant contribution of the majority of these traits in discriminating yam accessions.

The hierarchical clustering revealed genetic similarities among landraces accessions that were grouped in the same cluster. Clustering of *D. rotundata*, *D. praehensilis*, and *D. cayenensis* accessions in clusters two and four corroborates the findings of Scarcelli et al. [38], who

reported *D. praehensilis* as the progenitor of *D. rotundata*. This also suggests the existence of a possible genetic relationship between *D. rotundata* and *D. cayenensis*. Many studies have supported this theory [46,47] and, as such, called it the *D. cayenensis-rotundata* complex. From the clustering, *D. alata* showed characteristics for high tuber yield, high setts multiplication ratio, higher number of tubers per plot, larger tuber size, low tuber oxidative browning, high leaf density, moderate resistance to YMV severity, and early maturity. *D. rotundata*, *D. cayenensis*, and *D. praehensilis* showed characteristics for high yield, high dry matter content, large tuber size, better resistance to YAD severity, high leaf density, high plant vigor, and late maturity. *D. bulbifera* and *D. dumetorum* showed characteristics for a higher number of tubers per plot, leaf density, plant vigor, and early maturity.

### 4.4. Traits Prediction (Indirect Selection) for Yield and Tuber Quality Attributes

One of the challenges in yam improvement is long growing cycles of the genotypes. Thus, any means to select for improving yield and good tuber quality characteristics in yam accessions using agronomic characteristics (indirect selection) will be of advantage. Our study suggests that low YMV severity, leaf density, tuber size, and setts multiplication ratio predict tuber yield, while tuber size, plant vigor, YMV severity, and senescence class predict tuber dry matter content. Of the observed predictor traits, tuber size and YMV severity predict both tuber yield and tuber dry matter content.

## 5. Conclusions

This study explored a panel of 191 yam accessions within six *Dioscorea* species for the identification of superior genotypes for farmers' and consumers' preferred traits (tuber yield, tuber dry matter, YMV severity, YAD severity, and tuber flesh oxidative browning). We observed variations in the performance of the landrace species with respect to all the agronomic and tuber quality traits assessed in the study. All the assessed parameters have the moderate to high heritability necessary for response to selection. We observed significant relationships among the assessed traits and paths, and coefficient analysis revealed predictor traits for indirect selection. Four cluster groupings with contrasting characteristics were also identified. Our study identified 20 stable landrace accessions within three *Dioscorea* spp. with above-average check performance for farmers' and consumers' preferred traits. These accessions could be advised to farmers, as well as considered in future yam improvement programs in DR Congo. Further characterization of these landraces is required with high throughput molecular markers to ascertain their genetic uniqueness before incorporation into future breeding programs. This will provide more insight into the challenge of linguistic polymorphism and the genetic diversity of these species for effective use as source of genetic reservoirs for yam improvement in DR Congo.

**Supplementary Materials:** The following are available online at https://www.mdpi.com/article/10.3390/agriculture12050599/s1, Table S1: List of the landrace accessions with better performance over checks mean for most preferred farmers and consumer traits and their stability.

**Author Contributions:** Conceptualization, I.I.A., D.O.O., P.A.A., J.G.A., and J.-C.L.M.; Methodology, I.I.A. and P.A.A.; Data analysis, I.I.A. and P.A.A.; Supervision, D.O.O., P.A.A., and J.G.A.; Writing—original draft, I.I.A. and P.A.A.; Writing—review and editing, I.I.A., P.A.A., D.O.O.., J.G.A., J.-C.L.M., I.M.C., and J.L.K. All authors have read and agreed to the published version of the manuscript.

**Funding:** The African trans-regional cooperation, through the Mobilité Université en Afrique (MOUNAF) project funded by the European Union Commission within the framework of the Intra-Africa Academic Mobility Scheme, granted a Ph.D. scholarship to the first author to study at the University of Kisangani, Congo. This study is also partially supported by the BMGF, and publication fees will be covered by the BMGF.

**Institutional Review Board Statement:** Not applicable.

**Informed Consent Statement:** Not applicable.

**Data Availability Statement:** Data used are available within this manuscript, including the supplementary files. Data can be obtained upon request from the corresponding author.

**Acknowledgments:** The authors acknowledge the provision of research funds to the first author by the MOUNAF project. The Directorate of Research and Finance office of the University of Kisangani is also acknowledged for managing the MOUNAF project. We appreciate the guidance the inspector of the Inspection Provinciale de l'Agriculture and the cooperation of all yam local farmers, local aids, and authorities that facilitated germplasm collection at the six territories used for the study. We also thank all other Ph.D. colleagues within the project and the department for their motivational support.

**Conflicts of Interest:** The authors declare no conflict of interest.

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
