# Peer review of "Assessment of the Yam Landraces (Dioscorea spp.) of DR Congo for Reactions to Pathological Diseases, Yield Potential, and Tuber Quality Characteristics"

_agriculture, doi:10.3390/agriculture12050599_

Round 1

Reviewer 1 Report

The research article by Adejumobi et al. focuses on identifying landraces (cultivated and wild species) having a superior performance for yam foliar diseases, agronomic and tuber quality traits. Overall, the paper is very well organized and has sound information. However, I do have some concerns before its final acceptance. (i) In the present study, the phenotypic data was collected only for two environments, but for complex quantitative traits phenotyping must be conducted in at least three independent test environments (e.g. three locations and/or years for field trials). (ii) Figure 2, the name of genotypes was not clear, improving the quality. (iii) Need improvement in language and grammar.

Author Response

We have provided a response to reviewer 1 in the attached document 

Reviewer 2 Report

The manuscript entitled Assessment of yam landraces (Dioscorea spp.) of DR Congo for reaction to pathological diseases, yield potential and tuber 
quality characteristics is a routine study conducted to solve a location specific problem and biotic stress.

The Introduction needs improvement , add latest literature from last two years about the status and conservation and aplication of Yam land races, their disease resistance and status of disease severity in the region.

landraces are usually low yielding and lack traits important for intensified agriculture , discuss the point in comparison to the core set used by your group in this study with proper justification.

have you taken into consideration the impact of global climate change or abberrations in temperature since last 5-10 years and its effect on you adaptation of land races 

for AUDPC evaluation to get an estimate of disease pressure  was it done under control or natural conditions, and was the area a hot spot of this disease , how did you manage the disease severity .

Can u throw some more light on the performance of landraces as compared to standard checks, how landraces were superior 

Have these core set been evaluated earlier for Distinctness Uniformity and Stability traits which are more important for morpohological characteriszation programme 

Author Response

We have provided a response to reviewer 2 comments and suggestions  in the attached document 

Reviewer 3 Report

In this research article, Adejumobi et al analyzed a population of Yam landraces and measured various traits that could be relevant for breeding programs including tuber size, tuber browning and resistance to foliar diseases. After analyzing the variance among the traits measures and clustering different groups, the authors draw correlations and causal relations between traits. Finally they identify accessions that perform better than check genotypes.

Overall I think that the measurements and analyses in this article are thoroughly performed and clearly presented. I have few minor comments:

- It would help the reader and improve the story line to include an introductory sentence at the beginning of each paragraph to explain why a specific analysis is conducted (ANOVA, clustering, path analysis, comparison with standard check controls).

- Figure 1 is repeated twice

- In table 4, spell out what TDa, TDb… stands for

- In Figure 1 (anthracnose severity phenotypes), it would guide the reader to briefly explain what is seen on the picture. What are the criteria for scoring 1,2,3,4 or 5?

- l466: what is “linguistic polymorphism” in this context?

Author Response

We have provided a response to reviewer 3 comments and suggestions  in the attached document 

Round 2

Reviewer 2 Report

English language and presentation can be further improved 

Author Response

Thanks so much for reviewing our manuscript and suggesting English language editing. The manuscript has now been revised and improved.